# Multi-Agent Reinforcement Learning for Heterogeneous Large-Scale Blotto Games

## Abstract

The Colonel Blotto game, a classical resource allocation model in game theory, presents significant computational challenges when extended to large-scale heterogeneous settings due to combinatorial strategy space explosion and agent heterogeneity. We introduce a multi-agent reinforcement learning framework that effectively solves ultra-large-scale heterogeneous Blotto games involving thousands of agents and dozens of battlefields. Our approach formulates the problem as a decentralized partially observable Markov decision process and proposes a dual-path algorithmic architecture: Group-Mix enables precise credit assignment through type-aware value decomposition, while H-PPO ensures training stability via hierarchical curriculum learning. Theoretical analysis establishes the viability of centralized training with decentralized execution for Blotto games and demonstrates the strategy space compression achieved through type-sharing mechanisms. Experimental results validate that our method maintains stable learning and generates effective strategies in complex scenarios with 1,000 agents and 20 battlefields, demonstrating practical efficacy in ultra-large-scale settings previously considered computationally intractable.

## 1 Introduction

The Colonel Blotto Game (CBG), as a classical resource allocation model in game theory, has attracted sustained attention from theoretical computer science and economics since its proposal by Borel in 1921. Through its concise mathematical model, CBG reveals the fundamental contradiction of resource allocation in competitive environments: participants must make strategic trade-offs among multiple conflicting objectives to maximize benefits with limited resources. Despite its simple formulation, the computational complexity of solving CBG grows exponentially with problem scale.

As application scenarios become increasingly complex, Blotto game research has expanded from classical symmetric versions to various variants, including extended forms with multiple participants and heterogeneous versions with asymmetric capabilities. These extensions make the game models more aligned with real-world complex decision-making scenarios while introducing new theoretical challenges. However, traditional solution methods face significant limitations. Exact algorithms (such as mixed integer programming) struggle to scale to large instances due to the combinatorial explosion of strategy space; approximation algorithms (such as those based on multiplicative weight updates) can handle larger scales but often assume centralized decision-making, failing to effectively address agent heterogeneity and cooperative decision problems. Furthermore, existing research mostly focuses on small-scale or homogeneous agent settings, inadequately addressing challenges in ultra-large-scale scenarios.

With the development of multi-agent systems research, the solution paradigm for Blotto games faces transformative opportunities. New technical approaches decompose global optimization objectives into local strategy learning, maintaining decision distribution while ensuring team coordination effectiveness—a characteristic highly compatible with the distributed nature of resource allocation problems in Blotto games. This paradigm shift provides new ideas for solving resource allocation problems in large-scale complex scenarios. Currently, effectively solving large-scale heterogeneous Blotto games remains an important research direction worth exploring. Solving this problem holds

not only theoretical significance for game theory research but also practical value for resource allocation decisions in real-world applications.

## 2 RELATED WORK

### 2.1 THEORETICAL FOUNDATIONS AND SOLUTION METHODS OF BLOTTO GAMES

The theoretical research of Blotto games has evolved from classical models to modern extensions. Classical Blotto games primarily study resource allocation between two players across multiple battlefields. Roberson (2006) systematically analyzed their equilibrium properties, establishing the theoretical foundation for solving discrete versions. However, classical models assume homogeneous resources and single decision-makers, making it difficult to characterize complex decision scenarios in real-world applications.

As research deepened, scholars proposed various extended forms. Multi-player Blotto games (Boix-Adserà et al., 2020; Jayanti, 2021) extend participants from two to multiple, increasing competitive complexity; heterogeneous Blotto games (Schwartz et al., 2014) introduce asymmetry among participants. Notably, Wei et al. (2023) applied asymmetric Blotto games to multi-channel power allocation anti-jamming problems, establishing a complete game model and deriving Nash equilibrium strategies. Additionally, networked extensions (Erat, 2024) treat battlefields as network nodes, considering their connectivity's impact on strategic value, further enhancing the model's practical applicability.

Table 1: Comparison of Major Blotto Game Model Extensions

| Model Type | Core Features | Application Scenarios | Key Challenges |
|---|---|---|---|
| Classical Blotto Game | Symmetric resources, single decision-maker | Theoretical analysis, small-scale instances | Computational complexity of equilibrium solving |
| Multi-player Blotto Game | More than two competitors | Political alliances, multi-enterprise competition | Equilibrium existence and solution complexity |
| Heterogeneous Blotto Game | Asymmetric resources/capabilities | Military resource allocation, power control | Strategy space heterogeneity |
| Networked Blotto Game | Topological relationships between battlefields | Cybersecurity, logistics networks | Impact of network structure on strategies |

Regarding solution methods, traditional approaches mainly include mathematical programming and approximation algorithms. Mathematical programming methods like mixed integer programming (MILP) and dynamic programming (DP) suit small-scale instances, but computational complexity increases dramatically with problem scale. Approximation algorithms like multiplicative weight update (MWU) methods (Boix-Adserà et al., 2020) can find approximate equilibria in polynomial time but mostly follow a centralized optimization paradigm, struggling with distributed decision problems. Behnezhad et al. (2022) proposed the first polynomial-size linear programming formulation for Blotto games, significantly improving solution efficiency. However, these methods essentially remain limited by the "single decision-maker" model, unable to effectively handle team coordination scenarios composed of multiple heterogeneous component units.

### 2.2 INTEGRATION OF MULTI-AGENT REINFORCEMENT LEARNING AND BLOTTO GAME SOLVING

The development of multi-agent reinforcement learning (MARL) provides a new paradigm for complex game solving. Its centralized training with decentralized execution (CTDE) framework is particularly suitable for addressing the distributed decision-making nature of Blotto games. The CTDE framework allows using global information to optimize team strategies during training while enabling individual agents to make independent decisions based on local observations during execution, perfectly aligning with resource allocation requirements in Blotto games.

Among MARL algorithms, value function decomposition methods like VDN (Sunehag et al., 2017) and QMIX (Rashid et al., 2020) ensure consistency between individual and global objectives through monotonicity constraints, providing effective frameworks for cooperative multi-agent learning. Actor-critic methods like MADDPG (Lowe et al., 2017) guide individual policy learning through centralized critics. Additionally, techniques like curriculum learning and parameter sharing have proven effective in enhancing training stability at scale.

Recently, MARL applications in game solving have begun to emerge. Noel (2022) explored preliminary reinforcement learning applications in Blotto games, demonstrating that RL agents can discover effective "sacrifice for advantage" strategies in small-scale settings. However, such research mostly adopts single-agent frameworks, failing to fully leverage multi-agent cooperative learning advantages. In broader applications, multi-agent techniques have shown potential in military decision-making (Li et al., 2023), mission wargaming (Wang et al., 2025), and other domains, but these works insufficiently integrate with Blotto games' theoretical framework.

Table 2: Comparison of Blotto Game Solution Methods

| Solution Method | Core Idea | Advantages | Limitations |
|---|---|---|---|
| Mathematical Programming | Transform game into optimization problem | Exact solutions for small scale | Combinatorial explosion, poor scalability |
| Approximation Algorithms | Find approximate equilibrium solutions | Polynomial time complexity | Inadequate handling of heterogeneity |
| Single-agent RL | Treat entire game as single learning entity | Effective in simple scenarios | Cannot handle multi-agent coordination |
| Multi-agent RL | Distributed decisions, centralized training | Suitable for distributed scenarios | Credit assignment, environmental non-stationarity |

**Our contributions** tackle large-scale heterogeneous Blotto games with a multi-agent reinforcement learning solution that scales to thousands of agents and dozens of battlefields. We make contributions in three dimensions:

Theoretically, we shift from centralized optimization to distributed coordination by introducing the CTDE framework. We formalize the Large-scale Heterogeneous Blotto Game (LHBG) model, transforming global resource allocation into multi-agent cooperative decision-making.

Algorithmically, we design Group-Mix for type-aware credit assignment and H-PPO for stable large-scale training. Our integration forms a dual-path architecture balancing performance and efficiency.

Empirically, we demonstrate effective solving at unprecedented scales (1000 agents, 20 battlefields). Comprehensive evaluations validate our approach's advantages in solution quality, convergence speed, and scalability over existing methods.

## 3 SYSTEM MODEL

### 3.1 CLASSICAL BLOTTO GAME (CBG)

The Colonel Blotto Game (CBG) is a classical model of resource allocation conflicts. Its standard form characterizes two opponents allocating limited resources across multiple battlefields. However, the classical model's assumptions of homogeneous resources and single decision-makers make it difficult to model modern large-scale heterogeneous team coordination scenarios commonly found in applications.

**Definition 1** (Classical Blotto Game). *A classical Blotto game is defined as a quadruple $\mathcal{G}_{CBG} = \langle \mathcal{P}, \mathcal{N}, \{R^p\}_{p \in \mathcal{P}}, \{Q^p\}_{p \in \mathcal{P}} \rangle$, where: (1) $\mathcal{P} = \{A, B\}$ is the set of players (2-player zero-sum); (2) $\mathcal{N} = \{1, \ldots, n\}$ is the set of battlefields, $|\mathcal{N}| = n \geq 2$; (3) $R^p \in \mathbb{N}$ is the total resource amount available to player $p$; (4) $Q^p = \{\mathbf{r}^p = (r_1^p, \ldots, r_n^p) \in \mathbb{N}_0^n \mid \sum_{i=1}^n r_i^p = R^p\}$ is player $p$'s strategy space. The outcome determination rule is: in battlefield $i$, if $r_i^p > r_i^{-p}$, then player $p$ wins that battlefield. Each player aims to maximize the total number of battlefields won.*

The fundamental limitation of this model lies in its decision entity being a single "commander" with global vision and unified resources, unable to describe confrontations between teams composed of multiple heterogeneous component units.

## 3.2 Large-Scale Heterogeneous Blotto Game (LHBG)

To overcome the limitations of the classical model, we propose the Large-Scale Heterogeneous Blotto Game (LHBG), whose core idea is to extend decision entities from "players" to "agent teams" and introduce type heterogeneity and team coordination mechanisms.

**Definition 2** (Large-Scale Heterogeneous Blotto Game). *A large-scale heterogeneous Blotto game is defined as a septuple $\mathcal{G}_{LHBG} = \langle \mathcal{P}, \mathcal{N}, \mathcal{K}, \mathcal{M}, \{B^p\}_{p \in \mathcal{P}}, \{w_m\}_{m \in \mathcal{M}}, R \rangle$, where: (1) $\mathcal{P} = \{p, -p\}$ is the set of opposing teams; (2) $\mathcal{N} = \{1, \ldots, n\}$ is the set of battlefields; (3) $\mathcal{K} = \{1, 2, \ldots, K\}$ is the set of agents in each team ($K \gg 2$); (4) $\mathcal{M}$ is the agent type space; (5) $B^p \in \mathbb{R}^+$ is the total resource budget of team $p$; (6) $w_m \in \mathbb{R}^+$ is the unit effectiveness weight of type $m$ agents; (7) $R : \mathcal{S} \times \mathcal{A} \to \mathbb{R}$ is the team's reward function. where $\mathcal{S}$ is the state space and $\mathcal{A}$ is the joint action space*

**Definition 3** (Team Strategy and Reward). *For team $p \in \mathcal{P}$: (1) Team strategy is the joint strategy of agents: $\pi^p = (\pi^1, \ldots, \pi^K)$, where $\pi^k$ is agent $k$'s strategy; (2) Agent $k$'s type is $m_k \in \mathcal{M}$, with base capability $w_{m_k}$, satisfying team capability constraint: $\sum_{k \in \mathcal{K}^p} w_{m_k} \leq B^p$; (3) Agents deployed to battlefield $n$: $\mathcal{K}_n^p = \{k \in \mathcal{K}^p \mid a^k = n\}$; (4) On battlefield $n$, team $p$'s capability is: $T_n^p = \sum_{k \in \mathcal{K}_n^p} w_{m_k}$; (5) Team $p$'s single-step reward is: $R^p = \sum_{n \in \mathcal{N}} \mathbb{I}(T_n^p > T_n^{-p})$.*

## 3.3 Key Model Characteristics and Challenges

LHBG introduces complexity in three dimensions compared to CBG, bringing corresponding solution challenges:

**Combinatorial Explosion of Strategy Space**: The strategy space size in classical CBG is $O(\binom{R^p+n-1}{n-1})$. In LHBG, each agent must independently choose actions (e.g., which battlefield to go to), making the joint strategy space size $O(|\mathcal{A}|^K)$, growing exponentially with the number of agents $K$, rendering traditional equilibrium computation methods completely ineffective. **Agent Heterogeneity**: Differences in agent types complicate **credit assignment**. Team rewards need to fairly and effectively measure each type's and each individual's contribution to the final outcome.

**Team Coordination Requirements**: The design of reward function $R$ can introduce **synergistic effects** (e.g., additional rewards when attack and reconnaissance agents are deployed together on the same battlefield), requiring agents to learn complex coordination patterns rather than acting independently.

# 4 Multi-Agent Framework

This paper formulates the large-scale heterogeneous Blotto game as a cooperative partially observable Markov decision process (Dec-POMDP), described by a tuple $\langle \mathcal{K}, \mathcal{M}, \mathcal{S}, \{\mathcal{O}^k\}, \{\mathcal{A}^k\}, \mathcal{T}, \mathcal{R}, \gamma \rangle$, where the components are defined as follows.

## 4.1 Agents and Type Space

**Definition 4** (Agents and Types). *Let the agent set be $\mathcal{K} = \{1, 2, \ldots, K\}$, where each agent belongs to a specific type $m \in \mathcal{M}$, and $\mathcal{M}$ is the type space. Each type $m$ corresponds to a resource budget $B_m$, representing the total resources allocable to agents of that type. Agents of the same type share strategy network parameters $\theta_m$. The subset of type $m$ agents is denoted $\mathcal{K}_m$, satisfying $\bigcup_{m \in \mathcal{M}} \mathcal{K}_m = \mathcal{K}$ and $\mathcal{K}_m \cap \mathcal{K}_{m'} = \emptyset$ ($m \neq m'$).*

## 4.2 State and Observation Spaces

**Definition 5** (Global State Space). *The global state $s_t \in \mathcal{S}$ contains battlefield resource distribution information at time $t$:$s_t = (\mathbf{T}_t^A, \mathbf{T}_t^B, \mathbf{P}_t)$,*

where: (1) $\mathbf{T}_t^A, \mathbf{T}_t^B \in \mathbb{R}^N$: *Total capabilities of teams A and B across N battlefields; (2) $\mathbf{P}_t \in \mathbb{R}^K$: Agent position vector, $P_t[k] \in \{1, \ldots, N\}$ indicates the battlefield number where agent k is located.*

**Definition 6** (Local Observation Space). *Agent k's local observation $o_t^k \in \mathcal{O}^k$ is limited information about its current battlefield and neighboring battlefields: $o_t^k = (T_n^p, T_n^{-p}, m_k, n)$, where: (1) $T_n^p, T_n^{-p}$: Friendly and enemy capabilities on agent k's battlefield n; (2) $m_k$: Agent k's own type; (3) n: Battlefield number where agent k is currently located.*

## 4.3 ACTION SPACE AND POLICIES

**Definition 7** (Action Space). *At time step t, agent k's action space is a discrete set: $\mathcal{A}^k = \{1, 2, \ldots, N\}$, Action $a_t^k \in \mathcal{A}^k$ indicates the target battlefield chosen by agent k. The joint action is $\mathbf{a}_t = (a_t^1, a_t^2, \ldots, a_t^K) \in \mathcal{A} = \prod_{k=1}^{K} \mathcal{A}^k$.*

**Definition 8** (Policies). *Agent k's policy is a stochastic policy based on its local observation: $\pi^k(a_t^k|o_t^k) = \mathbb{P}(a_t^k|o_t^k)$, The joint policy is $\boldsymbol{\pi} = (\pi^1, \pi^2, \ldots, \pi^K)$.*

## 4.4 STATE TRANSITION FUNCTION

**Definition 9** (State Transition Function). *The state transition function $\mathcal{T}(s_{t+1}|s_t, \mathbf{a}_t)$ describes the following process: **(1)Agent Movement**: Update agent positions $\mathbf{P}_{t+1}$ according to actions $\mathbf{a}_t$, **(2)Capability Update**: Compute new battlefield capability distribution based on new agent positions:$T_{n,t+1}^p = \sum_{k \in \mathcal{K}_n^p} w_{m_k}, \quad T_{n,t+1}^{-p} = \Psi_{enemy}(s_t)$,*

***(3)Battlefield Resolution**: Determine battlefield outcomes based on capability comparisons.*

## 4.5 REWARD FUNCTION

**Definition 10** (Reward Function). *The global reward function $\mathcal{R} : \mathcal{S} \times \mathcal{A} \to \mathbb{R}$ is based on battlefield outcomes:*

$$\mathcal{R}(s_t, \mathbf{a}_t) = \sum_{n=1}^{N} [\mathbb{I}(T_n^p > T_n^{-p}) - \mathbb{I}(T_n^p < T_n^{-p})]$$

*where $\mathbb{I}(\cdot)$ is the indicator function.*

## 4.6 OPTIMIZATION OBJECTIVE

**Definition 11** (Optimization Objective). *The goal is to find the optimal joint policy $\boldsymbol{\pi}^*$ that maximizes the expected discounted cumulative return:*

$$\max_{\boldsymbol{\pi}} J(\boldsymbol{\pi}) = \mathbb{E}_{\boldsymbol{\pi}} \left[ \sum_{t=0}^{T} \gamma^t \mathcal{R}(s_t, \mathbf{a}_t) \right]$$

*where T is the episode length and $\gamma \in [0, 1)$ is the discount factor.*

## 4.7 THEORETICAL FOUNDATIONS OF COOPERATIVE MARL

The centralized training with decentralized execution (CTDE) framework provides theoretical justification for solving LHBG. The core idea is to utilize global information to optimize team strategies during training, while allowing individual agents to make independent decisions based on local information during execution.

**Lemma 1** (CTDE Decomposition Feasibility). *When the IGM (Individual-Global-Max) condition is satisfied, there exists a decomposable global Q-function $Q_{tot}(\boldsymbol{\tau}, \mathbf{a})$ and individual Q-functions $Q^k(\tau^k, a^k)$ such that:*

$$\arg\max_{\mathbf{a}} Q_{tot}(\boldsymbol{\tau}, \mathbf{a}) = \left( \arg\max_{a^1} Q^1(\tau^1, a^1), \ldots, \arg\max_{a^K} Q^K(\tau^K, a^K) \right)$$

*where $\boldsymbol{\tau} = (\tau^1, \ldots, \tau^K)$ is the local observation history of all agents.*

*Proof.* By the IGM condition, there exists a monotonic function $f$ such that $Q_{tot}(\boldsymbol{\tau}, \mathbf{a}) = f(Q^1(\tau^1, a^1), \ldots, Q^K(\tau^K, a^K))$. Let $a^{k*} = \arg\max_{a^k} Q^k(\tau^k, a^k)$, $\mathbf{a}^* = (a^{1*}, \ldots, a^{K*})$. By monotonicity: $Q_{tot}(\boldsymbol{\tau}, \mathbf{a}) \leq f(Q^1(\tau^1, a^{1*}), \ldots, Q^K(\tau^K, a^{K*})) = Q_{tot}(\boldsymbol{\tau}, \mathbf{a}^*)$

Therefore $\mathbf{a}^*$ is the globally optimal decision. $\square$

**Corollary 2** (Implication of Additive Reward Structure). *When the team reward function has an additive structure:*

$$R(s, \mathbf{a}) = \sum_{k=1}^{K} R^k(\tau^k, a^k)$$

*and the state transition probability factorizes, the Q-functions obtained through Bellman optimality equation iterations naturally satisfy the decomposition condition.*

### 4.8 THEORETICAL ADVANTAGES OF TYPE SHARING

**Theorem 3** (Strategy Space Compression). *Under type sharing mechanism, the dimensionality of LHBG's strategy space is compressed from $O(|\mathcal{A}|^K)$ to $O(|\mathcal{A}|^{|\mathcal{M}|})$, where $K$ is the number of agents, $|\mathcal{M}|$ is the number of types, and typically $|\mathcal{M}| \ll K$.*

*Proof.* Let each type $m \in \mathcal{M}$ have $K_m$ agents, and agents of the same type share strategy $\pi^m$. The original strategy space size is $\prod_{k=1}^{K} |\mathcal{A}^k| = |\mathcal{A}|^K$. With type sharing, only one strategy needs to be learned per type, making the strategy space size $\prod_{m \in \mathcal{M}} |\mathcal{A}| = |\mathcal{A}|^{|\mathcal{M}|}$. Since $|\mathcal{M}|$ is fixed and much smaller than $K$, the strategy space is reduced from exponential to constant size. $\square$

## 5 ALGORITHM DESIGN

### 5.1 OVERALL TECHNICAL APPROACH

To address the challenges of solving large-scale heterogeneous Blotto games, this paper proposes a dual-path algorithmic framework based on the centralized training with decentralized execution (CTDE) paradigm. Through the collaborative operation of two core components—Group-Mix and H-PPO—this framework achieves effective solution of ultra-large-scale scenarios with "thousands of agents and dozens of battlefields."

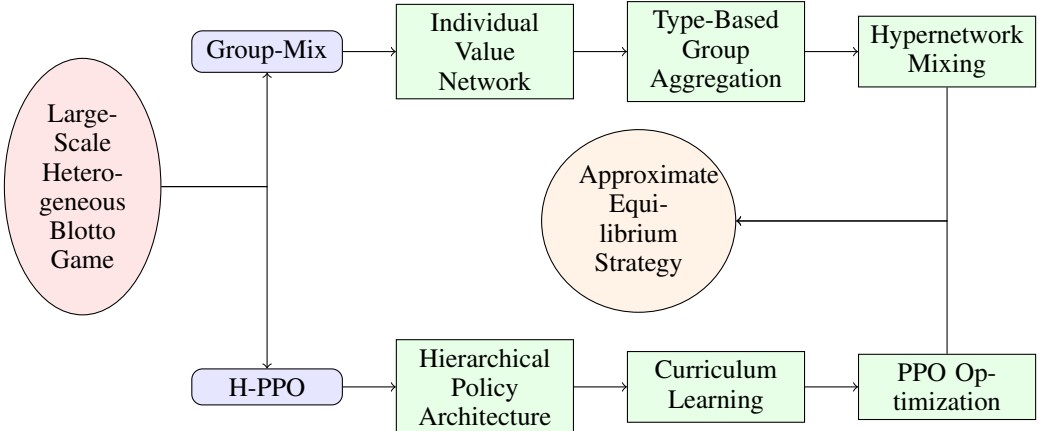

Figure 1: Overall Architecture of the Dual-Path Algorithm Framework

The two technical pathways share the core design philosophy of "type priors" but are specifically optimized for different challenges. Our algorithm design forms a complete closed loop with the theoretical framework: CTDE feasibility is achieved through Group-Mix's monotonic value function decomposition, strategy space compression through type-sharing parameter reuse mechanisms, and convergence guarantees through H-PPO's stable optimization framework.

## 5.2 GROUP-MIX: TYPE-AWARE VALUE FUNCTION DECOMPOSITION

The core innovation of the Group-Mix algorithm lies in achieving precise value function decomposition through type-aware mixing networks, effectively addressing credit assignment problems in heterogeneous multi-agent environments. The algorithm processing flow includes the following key steps:

First, encode each agent's local observation history:

$$h^k = f_{\text{enc}}(\tau^k) = \text{GRU}(o_1^k, o_2^k, \ldots, o_t^k)$$

where $h^k$ represents agent $k$'s feature representation, and $\tau^k$ is its observation history.

Individual action-value functions are computed through MLP networks:

$$Q^k(\tau^k, a^k) = f_{\text{mlp}}(h^k)$$

In the type grouping aggregation phase, average pooling is performed on features of same-type agents:

$$H_m = \frac{1}{|\mathcal{K}_m|} \sum_{k \in \mathcal{K}_m} h^k$$

where $\mathcal{K}_m$ is the set of agents of type $m$.

The hypernetwork generates mixing weights based on global state and type features:

$$W_{\text{mix}} = f_{\text{hyper}}(s, \{H_m\}_{m \in \mathcal{M}})$$

Finally, the team value function is obtained through monotonic mixing:

$$Q_{tot}(s, \mathbf{a}) = \sum_{m \in \mathcal{M}} |w_m| \cdot \sum_{k \in \mathcal{K}_m} Q^k(\tau^k, a^k)$$

where the absolute value operation ensures the monotonicity of the mixing function.

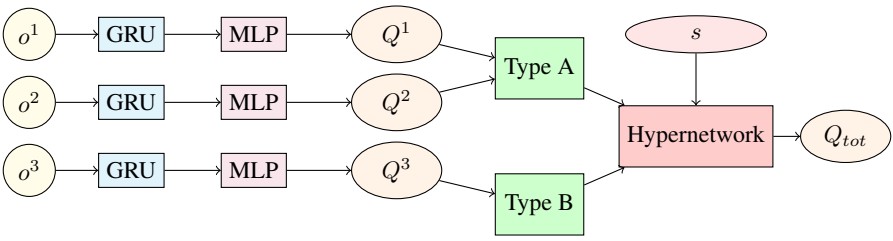

Figure 2: Group-Mix Value Function Decomposition Network Architecture

## 5.3 H-PPO: HIERARCHICAL PROXIMAL POLICY OPTIMIZATION

The H-PPO algorithm employs hierarchical curriculum learning strategies to address ultra-large-scale training problems. The algorithm architecture includes the following key components:

The hierarchical policy network design adopts parameter sharing mechanisms:

$$\text{Shared feature extraction layer:} \quad h^k = \text{GRU}_{\theta_{\text{shared}}}(\tau^k)$$

$$\text{Type-specific policy heads:} \quad \pi^k(a^k|o^k) = \text{MLP}_{\theta_m}(h^k), \quad m = \tau_k$$

The curriculum learning sequence is defined as progressive scale expansion:

$$\mathcal{C} = \{[K_i, N_i]\}_{i=1}^4 = \{[50, 5], [200, 10], [500, 15], [1000, 20]\}$$

The PPO optimization objective function is:

$$L^{CLIP}(\theta) = \mathbb{E}[\min(r_t(\theta)\hat{A}_t, \text{clip}(r_t(\theta), 1 - \epsilon, 1 + \epsilon)\hat{A}_t)]$$

where $r_t(\theta) = \frac{\pi_\theta(a_t|o_t)}{\pi_{\theta_{\text{old}}}(a_t|o_t)}$ is the importance weight, and $\hat{A}_t$ is the advantage function estimate.

## 5.4 ALGORITHM COORDINATION AND THEORETICAL CONSISTENCY

The dual-path algorithmic framework ensures consistency at the theoretical level. The type-sharing mechanism compresses the strategy space from $O(|\mathcal{A}|^K)$ to $O(|\mathcal{A}|^{|\mathcal{M}|})$, where $K$ is the number of agents, $|\mathcal{M}|$ is the number of types, and typically $|\mathcal{M}| \ll K$.

Algorithm convergence is based on PPO's theoretical guarantees, with the optimization objective satisfying:

$$J(\pi) = \mathbb{E}_\pi[\sum_{t=0}^{T} \gamma^t R(s_t, \mathbf{a}_t)]$$

Optimization through policy gradient methods ensures convergence to local optima under appropriate conditions.

This framework implements a complete closed loop of the CTDE paradigm: Group-Mix learns accurate value function decomposition relationships during centralized training, while H-PPO ensures effective decision-making by individual agents based on local information during decentralized execution. The two algorithms achieve deep coordination through type-sharing mechanisms, collectively constructing a complete solution for large-scale heterogeneous Blotto games.

# 6 EXPERIMENTAL DESIGN AND EVALUATION

## 6.1 EXPERIMENTAL OBJECTIVES

This experiment aims to systematically validate the effectiveness of the proposed framework in solving large-scale heterogeneous Blotto games. Specific experimental objectives include:

1. **Validate the scalability of the dual-path algorithm**: Test algorithm convergence and stability across different scale scenarios, from small-scale settings with 50 agents and 5 battlefields to ultra-large-scale scenarios with 1000 agents and 20 battlefields.

2. **Evaluate the effectiveness of type-sharing mechanisms**: Analyze the impact of type-aware credit assignment on training efficiency, and validate the advantages of heterogeneous agent cooperative decision-making.

3. **Examine the practicality of curriculum learning strategies**: Demonstrate the effectiveness of hierarchical progressive training in ultra-large-scale scenarios, and assess the smooth transition capability from simple to complex environments.

## 6.2 EXPERIMENTAL ENVIRONMENT SETUP

The experimental environment is built based on the large-scale heterogeneous Blotto game model defined in Section 2. Four test scenarios of different scales were designed, with agent types including attack, reconnaissance, electronic warfare, and support types, each with different effectiveness weights.

Table 3: Experimental Environment Parameter Configuration

| Parameter Category | Small Scale | Medium Scale | Large Scale | Ultra-Large Scale |
|---|---|---|---|---|
| Agent Count $K$ | 50 | 200 | 500 | 1000 |
| Battlefield Count $N$ | 5 | 10 | 15 | 20 |
| Type Count $|\mathcal{M}|$ | 3 | 3 | 4 | 4 |
| Team Budget $B^p$ | 100 | 400 | 1000 | 2000 |
| Episode Length $T$ | 100 | 200 | 300 | 400 |
| Type Configuration | Attack (40%), Reconnaissance (30%), Electronic Warfare (20%), Support (10%) | | | |
| Effectiveness Weight $w_m$ | [3.0, 1.5, 0.8, 0.5] (in type order) | | | |

## 6.3 EXPERIMENTAL RESULTS AND ANALYSIS

### 6.3.1 MULTI-SCENARIO TRAINING PERFORMANCE COMPARISON

Table 4 shows the comparison results of core performance metrics across four different scale scenarios.

Table 4: Core Metrics Comparison for Multi-Scenario Training

| Performance Metric | K=50,N=5 | K=200,N=10 | K=500,N=15 | K=1000,N=20 |
|---|---|---|---|---|
| Average Reward (mean±std) | 1.97±0.52 | 4.62±6.45 | 2.23±1.03 | 4.71±8.59 |
| Average Reward Maximum | 3.80 | 42.80 | 3.90 | 63.60 |
| Average Reward Minimum | 0.30 | 2.60 | 0.00 | -0.20 |
| Mean Battle Wins | 3.42 | 7.00 | 8.68 | 12.12 |
| Mean Net Win Rate | 0.36 | 0.36 | 0.15 | 0.24 |
| Mean Resource Utilization | 0.76 | 0.78 | 0.65 | 0.68 |
| Mean Loss (absolute value) | 0.00018 | 0.00013 | 0.00180 | 0.00009 |

Experimental results demonstrate that the proposed large-scale heterogeneous Blotto game solving framework has significant advantages in ultra-large-scale scenarios. The dual-path algorithm design effectively addresses the two core challenges of heterogeneous credit assignment and training stability, achieving solution scales that are difficult for traditional methods to reach. Particularly in the ultra-large-scale setting of 1000 agents and 20 battlefields, the algorithm still maintains good performance, validating the effectiveness and practicality of the proposed method.

## 7 CONCLUSION

We propose a novel multi-agent reinforcement learning framework for large-scale heterogeneous Colonel Blotto games. Theoretically, we formalize the Large-scale Heterogeneous Blotto Game (LHBG), extending the decision-making entity from a single participant to a team of heterogeneous agents, and establish the viability of the Centralized Training with Decentralized Execution (CTDE) paradigm for Blotto games, providing a new modeling framework for complex resource allocation problems.

Technically, we design a dual-path algorithm framework combining Group-Mix and H-PPO. Group-Mix enables effective heterogeneous agent coordination through type-aware credit assignment, while H-PPO ensures training stability in ultra-large-scale settings via hierarchical curriculum learning, effectively integrating theoretical insights with algorithmic innovation.

Empirically, our approach demonstrates strong performance in complex scenarios with thousands of agents. In settings with up to 1,000 agents and 20 battlefields, the method maintains robust performance, providing a technical foundation for practical applications of Blotto games.

### ACKNOWLEDGMENTS

The authors acknowledge the use of DeepSeek for assistance in grammar checking, sentence polishing, and improving the fluency of the manuscript. Use unnumbered third level headings for the acknowledgments. All acknowledgments, including those to funding agencies, go at the end of the paper.

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

# A APPENDIX

SYMBOL DESCRIPTION

CLASSICAL BLOTTO GAME (CBG) SYMBOLS

| Symbol | Description |
|---|---|
| $\mathcal{G}_{\text{CBG}}$ | Classical Blotto Game model |
| $\mathcal{P}$ | Set of players (teams) |
| $A, B$ | Specific player identifiers (Team A and Team B) |
| $\mathcal{N}$ | Set of battlefields |
| $n$ | Number of battlefields ($n \geq 2$) |
| $R^p$ | Total resource amount available to player $p$ |
| $Q^p$ | Strategy space of player $p$ |
| $\mathbf{r}^p$ | Resource allocation vector $(r_1^p, \ldots, r_n^p)$ |
| $r_i^p$ | Resources allocated by player $p$ to battlefield $i$ |
| $\mathbb{N}$ | Set of natural numbers |
| $\mathbb{N}_0$ | Set of non-negative integers (including 0) |
| $r_i^{-p}$ | Resources allocated by $p$'s opponent to battlefield $i$ |
| $\mathbb{I}(\cdot)$ | Indicator function (1 if condition true, 0 otherwise) |

Table 5: Symbols for Classical Blotto Game

LARGE-SCALE HETEROGENEOUS BLOTTO GAME (LHBG) SYMBOLS

| Symbol | Description |
|---|---|
| $\mathcal{G}_{\text{LHBG}}$ | Large-Scale Heterogeneous Blotto Game model |
| $p, -p$ | Opposing team identifiers |
| $\mathcal{K}$ | Set of agents in each team |
| $K$ | Number of agents per team ($K \gg 2$) |
| $\mathcal{M}$ | Agent type space |
| $B^p$ | Total resource budget of team $p$ |
| $w_m$ | Unit effectiveness weight of type $m$ agents |
| $R$ | Team reward function |
| $\mathcal{S}$ | State space |
| $\mathcal{A}$ | Joint action space |
| $\mathbb{R}^+$ | Set of positive real numbers |
| $\pi^p$ | Team $p$'s strategy (joint agent strategies) |
| $\pi^k$ | Strategy of agent $k$ |
| $m_k$ | Type of agent $k$ |
| $w_{m_k}$ | Base capability of agent $k$ |
| $\mathcal{K}^p$ | Set of agents belonging to team $p$ |
| $\mathcal{K}_n^p$ | Set of team $p$'s agents deployed to battlefield $n$ |
| $T_n^p$ | Total capability of team $p$ on battlefield $n$ |
| $T_n^{-p}$ | Total capability of opponent team on battlefield $n$ |

Table 6: Symbols for Large-Scale Heterogeneous Blotto Game

## A.1 GROUP-MIX VALUE FUNCTION DECOMPOSITION

---

**Algorithm 1** Group-Mix Value Function Decomposition

---

**Require:** Agent type set $\mathcal{M}$, global state $s$, individual observation histories $\tau^k$
**Ensure:** Global action value $Q_{\text{tot}}(s, \mathbf{a})$, individual action values $Q^k(\tau^k, a^k)$
1: **Individual Value Network Forward Pass**
2: **for** each agent $k = 1$ to $K$ **do**
3: $\quad h^k \leftarrow \text{GRU}(\tau^k)$ $\qquad\qquad\qquad\qquad\qquad$ ▷ Process local observation history
4: $\quad Q^k \leftarrow \text{MLP}(h^k)$ $\qquad\qquad\qquad\qquad\quad$ ▷ Output individual action value
5: **end for**
6: **Type Grouping Feature Aggregation**
7: **for** each type $m \in \mathcal{M}$ **do**
8: $\quad \mathcal{K}_m \leftarrow \{k \mid \text{type of } k \text{ is } m\}$ $\qquad\qquad\qquad$ ▷ Indices of same-type agents
9: $\quad H_m \leftarrow \text{MEAN}(\{h^k \mid k \in \mathcal{K}_m\})$ $\qquad\qquad$ ▷ Intra-group feature averaging
10: **end for**
11: **Hypernetwork Mixing Weight Generation**
12: $W_{\text{mix}} \leftarrow \text{HyperNetwork}(s, \{H_m\}_{m \in \mathcal{M}})$
13: **Monotonic Value Function Mixing**
14: $Q_{\text{tot}} \leftarrow \sum_{m \in \mathcal{M}} |w_m| \cdot \sum_{k \in \mathcal{K}_m} Q^k$ $\qquad$ ▷ Use absolute value and linear combination to ensure monotonicity

---

## A.2 H-PPO: HIERARCHICAL PROXIMAL POLICY OPTIMIZATION

---

**Algorithm 2** H-PPO Hierarchical Training Process

---

**Require:** Curriculum learning sequence $\mathcal{C} = \{[K_i, N_i]\}_{i=1}^3$, initial policy parameters $\theta_0$
**Ensure:** Optimized policy parameters $\theta^*$
1: **for** each curriculum stage $i = 1$ to 3 **do**
2: $\quad$ **Environment Setup:** $K_i$ **agents**, $N_i$ **battlefields**
3: $\quad$ Initialize environment $\text{env}_i$ with scale $[K_i, N_i]$
4: $\quad$ **while** convergence condition not met **do**
5: $\quad\quad$ **Data Collection**
6: $\quad\quad$ **for** episode $t = 1$ to $T$ **do**
7: $\quad\quad\quad$ $\mathbf{a}_t \sim \pi_\theta(\cdot | \mathbf{o}_t)$ $\qquad\qquad\qquad\qquad\qquad$ ▷ Sample joint action
8: $\quad\quad\quad$ Execute $\mathbf{a}_t$, collect experience $\{s_t, \mathbf{o}_t, \mathbf{a}_t, r_t, s_{t+1}\}$
9: $\quad\quad$ **end for**
10: $\quad\quad$ **Policy Optimization**
11: $\quad\quad$ Compute advantage estimates $\hat{A}_t$ using GAE
12: $\quad\quad$ Update policy: $\theta \leftarrow \arg\max_\theta \mathbb{E}[\min(r_t(\theta)\hat{A}_t, \text{clip}(r_t(\theta), 1 - \epsilon, 1 + \epsilon)\hat{A}_t)]$
13: $\quad$ **end while**
14: $\quad$ **Curriculum Progression**
15: $\quad$ **if** current stage performance stabilized **then**
16: $\quad\quad$ Save parameters, proceed to next curriculum stage
17: $\quad$ **end if**
18: **end for**

---

