# OpenReview forum: "Multi-Agent Reinforcement Learning for Heterogeneous Large-Scale Blotto Games"
_ICLR.cc/2026/Conference — Submitted to ICLR 2026_

### Official Review · Reviewer_LRob · 2025-10-28

**Soundness:** 1
**Presentation:** 2
**Contribution:** 1
**Rating:** 2
**Confidence:** 4

**Summary:**

This paper applies multi-agent reinforcement learning algorithms under the CTDE paradigm to large-scale heterogeneous Colonel Blotto game problems by designing a dual-path algorithmic framework combining Group-Mix and H-PPO. To validate this approach, the paper develops reinforcement learning environments for Colonel Blotto games at various scales. Experiments across different scenarios demonstrate that the proposed method exhibits competitive performance in complex, large-scale settings.

**Strengths:**

1. The paper introduces and proposes a multi-agent reinforcement learning environment for the large-scale heterogeneous Colonel Blotto game, which contributes to solving the Colonel Blotto problem.
2. The paper successfully applies multi-agent reinforcement learning algorithms to the Colonel Blotto game and proposes two methods: Group-Mix and H-PPO.

**Weaknesses:**

1. The novelty of the paper is limited. The distinctions between Group-Mix and QMIX, as well as between H-PPO and PPO, are not clearly articulated. Furthermore, the convergence proof presented does not seem to be an original contribution of this paper.
2. The experimental section lacks critical details necessary for reproducibility. This includes the specific hyperparameter settings, the number of random seeds used for statistical significance, a complete list of baseline methods for comparison, and a thorough ablation study.
3. The paper should introduce a more comprehensive introduction to the proposed Colonel Blotto game benchmark. Key details are missing, such as the performance of classical algorithms on this benchmark and a discussion of possible environment variants.

**Questions:**

1. Could the authors clarify the primary distinctions between the "single decision-maker" model and multi decision maker model discussed in the paper? Furthermore, is the proposed method capable of solving traditional "single decision-maker" problems?
2. It would be valuable to see a comparative analysis of the algorithm's performance and computational efficiency against other baseline methods, such as QMIX, MAPPO, and MADDPG.
3. Could the authors discuss whether single-agent algorithms (e.g., DQN, PPO) can be directly applied to the multi-agent reinforcement learning environment proposed in this work.

---

### Official Review · Reviewer_W453 · 2025-10-31

**Soundness:** 2
**Presentation:** 2
**Contribution:** 2
**Rating:** 4
**Confidence:** 3

**Summary:**

This paper investigates a solution for heterogeneous larg-scale blotto games. Approaching the problem as a decentralized POMDP problem, this paper incorporates ideas such as Group-Mix, H-PPO, and hierarchical curriculum learning. The proposed approach was tested in scenarios up to 1K agents and 20 battlefields.

**Strengths:**

This paper introduces the Blotto games quite well. It also has a good characterization of challenges involved and previous approaches to the problem.

**Weaknesses:**

Overall, I found the description of the algorithm rather confusion. For example,What's the Relationship between GRU_theta_shared and MLP_theta_m? Is the Q values calculated from the Group-Mix Value Function Decomposition used for training of the H-PPO? Where does two pathways merge and interact with each other to make a decision? While the Figure 2 was helpful, Figure 1 was more confusing than it helped me to understand the algorithm. I also think that having a pseudocode algorithm would be helpful too.

In this regard, I think there's some room for improvement in the figures. While not critical, it's quite better to change lines if a word does not fit into a box in the figures. Center-aligning also helps since letters too close to box outlines can be difficult to read. Ideally, best figures are figures that so clearly describes the proposed approach that the reader should have a grasp of what the paper is about, even if they saw the figures before reading through the whole text.

I also think the experiments are bit incomplete. It is nice that the authors tested their algorithm on various scale of Blotto games to get a sense on scalability. However, there are no comparison with other state-of-the-art algorithms in the field, so it is difficult to measure the contribution of this approach.

**Questions:**

1. A better description of the algorithm. Currently, it is difficult to see the structural / theoretical contribution of the algorithm as it is difficult to understand how the algorithm is structured and works

2. Comparison with the baseline to better understand the contribution of this work to existing works of Colonel Blotto Games.

---

### Official Review · Reviewer_w1yK · 2025-11-01

**Soundness:** 1
**Presentation:** 1
**Contribution:** 2
**Rating:** 0
**Confidence:** 4

**Summary:**

This paper approaches solving large-scale heterogeneous Colonel Blotto games using a centralized training, decentralized execution framework. Using their proposed Group-Mix and Hierarchical PPO method, they claim strong performance on large versions of the game.

**Strengths:**

The authors consider solving large-scale extensions of Blotto.

**Weaknesses:**

This paper holds very little resemblance to research works typically accepted to major machine learning conferences.

- The introduction doesn't introduce the contributions or the proposed approach. The actual purpose of the paper is introduced after related works. This structure seems very odd and should be streamlined.
- The related work section provides only high-level positioning. It does not clearly spell out how prior approaches address (or fail to address) the considered problem, nor how the proposed method differs from and builds on them.
- Although core methods are cited, the paper doesn’t make explicit which components of the proposed approach derive from which prior works or how they are adapted.
- There is a single experiment with zero ablations.
- The experimental analysis is extremely insufficient, being a single paragraph.
- There are no baselines for the proposed algorithm. No point of reference for whether this algorithm solves something better or worse than other methods.
- There are no training details provided.
- The acknowledgement section still contains the template paper instructions.

**Questions:**

What is a baseline method that could be used to measure the relative performance of this method to prior work? I suggest comparing to a baseline to help readers understand when and why your method is preferable to prior work.

---

### Official Review · Reviewer_3V1X · 2025-11-01

**Soundness:** 2
**Presentation:** 3
**Contribution:** 2
**Rating:** 2
**Confidence:** 4

**Summary:**

This paper studies large-scale, heterogeneous Colonel Blotto games by casting them as cooperative Dec-POMDPs and proposing a dual-path MARL framework: Group-Mix for type-aware value decomposition and H-PPO for stable, scalable training. Theoretically, the authors argue CTDE applicability under an IGM-style monotonic factorization and claim a strategy-space compression from $O(|A|^K)$ to $O(|A|^{|M|})$ through type sharing. Empirically, they formalize the Large-scale Heterogeneous Blotto Game and runs up to 1,000 agents and 20 battlefields.

**Strengths:**

1. The paper clearly reformulates heterogeneous Blotto as a Dec-POMDP and enumerates the sources of difficulty (combinatorial joint action space, credit assignment under type heterogeneity, coordination effects). The LHBG definitions (agents, types, local observations, rewards) are easy to follow.
2. Group-Mix explicitly aggregates same-type features before hypernetwork mixing, aiming to align the mixer with heterogeneity structure and improve credit assignment, which is an intuitively appealing inductive bias for large populations.
3. The type-sharing mechanism (one policy/value head per type) is a sensible, principled way to reduce degrees of freedom and training variance for thousands of agents, and the paper articulates its impact on the strategy space.

**Weaknesses:**

1. Single environment: the current evaluation is conducted in a single environment, which may limit the perceived generality and external validity of the results. To better demonstrate robustness and transfer, it would be helpful to include widely used MARL benchmarks and diverse tasks—for example, several maps from SMAC [1], as well as scenarios from GRF [2] and Hanabi [3].
2. Unclear improvements over existing MARL algorithm: the paper would benefit from a clearer articulation of how the proposed method advances beyond established MARL approaches. In particular, classical baselines such as MAPPO [4] (centralized critic with decentralized policies) and QMIX [5] (monotonic value factorization) set strong reference points. Please make explicit what design choices are new relative to these methods, which concrete challenges they target and why those choices should help in theory or practice.
3. Limited experiments: at present, there are no baseline comparisons or ablation studies, which makes it difficult to judge effectiveness and attribute gains to specific design elements. Including competitive baselines (e.g., MAPPO, QMIX) and well-scoped ablations would substantially improve the evidential weight of the claims. What's more, there is no description of the network architecture, optimization details, or hyperparameters.

[1] Samvelyan, Mikayel, et al. "The starcraft multi-agent challenge." arXiv preprint arXiv:1902.04043 (2019).

[2] Kurach, Karol, et al. "Google research football: A novel reinforcement learning environment." Proceedings of the AAAI conference on artificial intelligence. Vol. 34. No. 04. 2020.

[3] Bard, Nolan, et al. "The hanabi challenge: A new frontier for ai research." Artificial Intelligence 280 (2020): 103216.

[4] Yu, Chao, et al. "The surprising effectiveness of ppo in cooperative multi-agent games." Advances in neural information processing systems 35 (2022): 24611-24624.

[5] Rashid, Tabish, et al. "Monotonic value function factorisation for deep multi-agent reinforcement learning." Journal of Machine Learning Research 21.178 (2020): 1-51.

**Questions:**

1. Evaluation environments: There are many established MARL environments; could the authors validate their method in a broader range of scenarios, such as Hanabi or SMAC? This would better demonstrate the generality and robustness of the proposed algorithm.
2. Baseline comparison: Could the authors compare their method with classical MARL algorithms such as MAPPO and QMIX to verify its effectiveness? Including such baselines would help clarify the algorithm’s relative advantages.
3. Experimental details: Could the authors provide more detailed descriptions of the experimental setup, including the network architecture and hyperparameter choices? These details are essential for reproducibility and fair comparison.
4. Opponent setup: Since the game involves two agents in an adversarial setting, what algorithm is used to control the opposing agent in the main experiments? Clarifying this is important for understanding the evaluation protocol and the fairness of the results.

---

### Meta-Review · Area_Chair_6p8d · 2026-01-05

**Summary:**

This paper attracted significant concerns from the reviewers, along all axes of evaluation. None of the reviewers indicated a positive score, and the authors did not post their rebuttals. No discussion was had between the reviewers and the authors.

My recommendation reflects the reviewers' consensus that the paper is not ready for publication.

**Reviewer Concerns:**

No rebuttal was posted.

**Reviewer Scores:**

No author-reviewers discussion was had.

---

### Decision · Program_Chairs · 2026-01-26

Reject